# Human Serum Albumin Nanoparticles: Synthesis, Optimization and Immobilization with Antituberculosis Drugs

**DOI:** 10.3390/polym15132774

**Published:** 2023-06-22

**Authors:** Aldana Galiyeva, Arailym Daribay, Tolkyn Zhumagaliyeva, Lyazzat Zhaparova, Daniyar Sadyrbekov, Yerkeblan Tazhbayev

**Affiliations:** Institute of Chemical Problems, Karagandy University of the Name of Academician E.A. Buketov, Karaganda City 100028, Kazakhstan; arailymdaribay@gmail.com (A.D.); zhumagalieva79@mail.ru (T.Z.); lyazzh@mail.ru (L.Z.); daniyar81@gmail.com (D.S.)

**Keywords:** nanoparticles, albumin, rifampicin, isoniazid, antituberculosis drugs, desolvation

## Abstract

The aim of this study was to create nanoparticles of human serum albumin immobilized with anti-TB drugs (rifampicin, isoniazid) using the desolvation method. Central Composite Design (CCD) was applied to study the effect of albumin, urea, L-cysteine, rifampicin and isoniazid concentration on particle size, polydispersity and loading degree of the drugs. The optimized nanoparticles were spherical in shape with an average particle size of 216.7 ± 3.7 nm and polydispersity of 0.286 ± 4.9. The loading degree of rifampicin and isoniazid in the optimized nanoparticles were 44% and 27%, respectively. The obtained nanoparticles were examined by Fourier-transform infrared spectroscopy (FTIR), thermogravimetric analysis (TGA) and differential scanning calorimetry (DSC); the results showed the absence of drug–polymer interactions. The drug release from the polymer matrix was studied using dialysis membranes.

## 1. Introduction

Tuberculosis (TB) is one of the world’s leading infectious killers. The COVID-19 pandemic continues to have a devastating impact on access to TB diagnosis and treatment. The progress up to 2019 has slowed, halted or reversed, and global TB targets are off track [1]. TB incidence rates (new cases per 100,000 population per year) increased by 3.6% between 2020 and 2021 [1]. The burden of drug-resistant TB is also estimated to increase, with 450,000 (95% UI: 399,000–501,000) new cases of rifampicin-resistant TB in 2020–2021 [1]. New approaches for the treatment of TB, including for multidrug-resistant TB, therefore need to be developed.

Isoniazid (INH) and rifampicin (RIF) are the two main anti-TB drugs widely used in the treatment of tuberculosis [2,3,4]. The combination of isoniazid and rifampicin supplements the action of each of the separately administered drugs and ensures more effective destruction of the bacteria. Moreover, the use of combination therapy reduces the likelihood of the bacteria developing resistance to the drugs [5,6]. However, it can also cause some side effects, such as hepatotoxicity, neurotoxicity, hypersensitivity and vitamin B6 deficiency [7,8,9]. It can also cause allergic reactions such as skin rashes, itching, urticaria, and sometimes more severe reactions such as angioedema, anaphylaxis and other allergic manifestations [10].

Reducing the side effects and resistance to rifampicin and isoniazid is a challenge, but several approaches exist, one of which is the use of polymer composites for drug transport [11,12,13,14]. The use of nanoparticles (NPs) may allow drugs to be delivered directly to the site of infection and reduce their toxicity to the body.

Biodegradable polymers such as polylactide (PLA), copolymers of polylactic and glycolic acids (PLGA), human (HSA) and bovine albumin, and polyethylene glycol are used for nanoparticle synthesis [15,16]. Compared to PLA and PLGA, albumin is water-soluble and easily processed; in addition, the albumin macromolecule contains different functional groups, so it can attach both positively charged and negatively charged drugs. [17,18,19]. Albumin is a natural protein present in the human body, so albumin-based nanoparticles have high biocompatibility. This reduces the possibility of immune reactions and toxicity, which is an important factor in the development of drug nanoparticles [17]. The use of albumin-based nanoparticles for the transport of TB drugs is promising, as bioavailability is improved, stability is increased, toxicity is reduced, transport specificity is increased, and resistance to drug resistance is improved [20,21,22].

For the synthesis of albumin-based nanoparticles, the desolvation method is the most commonly used [23,24,25]. This method allows the synthesis of nanoparticles by a simple process of adding desolvating agents, such as ethanol and acetone, to albumin solutions containing drugs. Desolvating agents change the structure of albumin and reduce its solubility, leading to the formation of precipitates in the form of protein nanoparticles [23]. Once the nanoparticles are formed, they are bonded by linking agents such as glutaric aldehyde [26]. In our work, we propose a method that avoids the use of a synthetic stabilizer and instead replaces it with natural agents such as urea and cysteine. This may open new perspectives for improving drug delivery and overcoming the toxicity associated with the use of synthetic stabilizers.

Albumin-based nanoparticles loaded with isoniazid and rifampicin represent a promising approach for the treatment of tuberculosis. However, determining the optimum conditions for the production of such nanoparticles is a challenging task requiring careful research and optimization of various process parameters. In this context, the central composite design (CCD) method is an effective tool to optimize and improve the process of producing albumin-based nanoparticles loaded with isoniazid and rifampicin in order to improve their effectiveness and stability.

Thus, we were interested in using biodegradable polymeric nanoparticles immobilized with isoniazid and rifampicin with sustained action and targeted delivery, with high efficacy and reduced toxic side effects to increase the effectiveness of tuberculosis treatment. Rifampicin- and isoniazid-loaded human serum albumin (HSA-INH-RIF) NPs were optimized by CCD and the effects of five factors were investigated: concentrations of HSA, urea, L-cysteine, rifampicin and isoniazid.

## 2. Materials and Methods

### 2.1. Materials

Human serum albumin (lyophilized powder, 98%), rifampicin and isoniazid with indicated purity over 99%, and L-cysteine (98.5%) were purchased from Sigma Aldrich (Taufkirchen, Germany). Absolute ethanol was purchased from DosFarm (Almaty, Kazakhstan). Urea (99.5%) was purchased from HimPribor-SPb (Saint Petersburg, Russia).

### 2.2. Production of Human Serum Albumin Nanoparticles Loaded with Isoniazid and Rifampicin by Desolvation

Nanoparticles of human serum albumin were prepared by desolvation [27]. According to this technique, a given amount of serum albumin was dissolved in distilled water while stirring at 200 rpm, avoiding clumping and foaming, at room temperature for 10 min. The concentration of these prepared protein solutions was 10–100 mg/mL. Then, a given amount of aqueous urea solution (at a concentration of 4–6 mol/L) was added and treated with ultrasound (Launch Tech, Shenzhen, China) for 3 min [28]. After that, 16 mL of ethanol was then added to each protein solution at a constant rate (1 mL/min), resulting in a turbid dispersion of albumin nanoparticles. An aqueous solution of L-cysteine was then added, in which the concentration of the amino acid was 0.1–2.5 mg/mL. Pre-prepared solutions of isoniazid and rifampicin (dissolved in an aqueous solution of DMSO) was added to the obtained HSA nanoparticles so that the drug concentration in the system was 2–8 mg/mL and the mixture was stirred (stirring speed was 300 rpm) for 2 h. The nanoparticle suspension was then purified by three cycles of centrifugation at 14,000 rpm (MiniSpin, Eppendorf, Hamburg, Germany) for 15 min to remove unabsorbed isoniazid and rifampicin.

### 2.3. Determination of Particle Size, Polydispersity and ζ Potential

The particle size and polydispersity of the nanoparticles were determined by photon correlation spectroscopy on a Zetasizer Nano S90 from Malvern (Malvern Instruments Ltd., Malvern, UK). For all measurements, each sample was diluted to the appropriate concentration with distilled water. Each dimensional assay lasted 120 s and was performed at 25 °C with a 90° angle determination. The ζ potential was measured with a ζ-potential analyzer (Zetasizer Nano ZS90, Malvern Instruments, Worcestershire, UK) using electrophoretic laser Doppler anemometry. In addition, the size, shape and surface morphology of the nanoparticles were investigated by scanning electron microscopy (MIRA 3LM TESCAN, Brno, Czech Republic, EU).

### 2.4. Determination of Loading Degree of Drugs and Nanoparticle Yield

The amount of drug in the supernatant was determined using high performance liquid chromatography (HPLC) (Shimadzu LC-20 Prominence). A volume of 0.1 mL of the supernatant collected after centrifugation was diluted with 3 mL of water and measured by HPLC at 254 nm. The mobile phase was water–acetonitrile–formic acid (94:5:1), with a flow rate of 1.5 mL/min. An Agilent 300 Extend (Agilent Technologies, Tokyo, Japan) C-18 column (sorbent grain size 5 μm, 100 Å, 4.6 × 150 mm) was used. The operating temperature of the column was maintained at 40 °C. The quantification method was internal area normalization. The instrument was set up for an injection volume of 10 μL (loop injection).

The amount of drug encapsulated in the polymer nanoparticles was determined by measuring the amount of unencapsulated drug in the aqueous solution recovered after ultracentrifugation and particle washing. The loading degree was calculated as follows:$$\mathrm{Loading}\;\mathrm{degree}\;\left(\%\right)=\frac{\mathrm{Mass}\;\mathrm{of}\;\mathrm{the}\;\mathrm{total}\;\mathrm{Drug}\;-\mathrm{Mass}\;\mathrm{of}\;\mathrm{free}\;\mathrm{Drug}}{\mathrm{Mass}\;\mathrm{of}\;\mathrm{total}\;\mathrm{nanoparticles}\;}\times 100\%$$
$$\mathrm{Nanoparticles}\;\mathrm{Yield}\;\left(\%\right)=\frac{\;\mathrm{Mass}\;\mathrm{of}\;\mathrm{total}\;\mathrm{nanoparticles}}{\mathrm{Mass}\;\mathrm{of}\;\mathrm{the}\;\mathrm{total}\;\mathrm{drug}+\mathrm{Mass}\;\mathrm{of}\;\mathrm{total}\;\mathrm{HSA}}\times 100\%$$

### 2.5. In Vitro Study of Drug Release from Polymer Nanoparticles

The cumulative drug release from the polymer matrix was determined by dialysis in phosphate-buffered saline (pH 7.4) at 37 °C. For this purpose, the nanoparticles immobilized by the drug were dispersed in phosphate-buffered saline and treated with ultrasound for 10 min. The resulting dispersion was transferred to a dialysis membrane (MWCO 8000D). The membrane, sealed with clamps, was placed in a dialysis beaker with buffer solution, covered with a lid and stirred on a magnetic stirrer at 200 rpm. Dialysate samples were taken at fixed time intervals such as 0.25, 0.5, 1, 2, 4, 8, 24 h. To study the degree of release from the polymer nanoparticles, the amount of drug released was recorded using HPLC and calculated according to the formula:$$\mathrm{Drug}\;\mathrm{release}\;\left(\%\right)=\frac{\;\mathrm{Mass}\;\mathrm{of}\;\mathrm{released}\;\mathrm{drug}}{\mathrm{Mass}\;\mathrm{of}\;\mathrm{the}\;\mathrm{total}\;\mathrm{drug}\;\mathrm{in}\;\mathrm{nanoparticles}}\times 100\%$$

### 2.6. Thermogravimetric Analysis and Differential Scanning Calorimetry

The polymer and nanoparticle behavior of PEG-BSA-INH during thermal degradation was studied with a thermogravimetric analyzer and differential scanning calorimetry (LabSYS evo TGA/DTA/DSC, Setaram, France); the instrument was operated from 30 °C to 600 °C under nitrogen at a flow rate of 30 mL/min and heating rate of 10 °C/min.

### 2.7. Study of Prepared Nanoparticles by Infrared Spectroscopy

IR spectroscopy (FSM 1202, Infraspek Ltd., Saint Petersburg, Russia) was used to identify the samples. The FT-IR spectra were determined using the KBr method. Approximately 3 mg of the sample was mixed with 100 mg of KBr and a pellet was prepared. The IR range investigated was 4000 to 400 cm^−1^ and the resolution of the FTIR spectra was 8 cm^−1^.

### 2.8. Experimental Design of Central Composite Design

The NPs were formulated according to a central composite design (CCD) to investigate the effect of the independent variables, i.e., the concentrations of has, urea, cysteine, isoniazid and rifampicin, on particle size and loading degrees of isoniazid and rifampicin. The CCD matrix was generated using Design Expert^®^ software (version 13, Stat-Ease, Minneapolis, MN, USA); to estimate five factors at five levels (Table 1), the design consisted of eleven factorial point batches, ten axial point batches and five replicates at central points.

## 3. Results

### 3.1. Optimization of Nanoparticles Preparation

The desolvation procedure is a common method to produce protein-based particles [29,30,31]. Desolvation of HSA with ethanol results in clearly delineated nanoparticles with denatured albumin forming a matrix of spheres [32]. Previous studies have mainly used glutardialdehyde as a crosslinking agent to prepare albumin-based nanoparticles [30]. We propose a method that eliminates the use of synthetic stabilizer, replacing it with natural agents such as urea and cysteine. For this purpose, we modified a previously developed technique for the anti-tumor drug hydroxyurea [33,34]. Human serum albumin nanoparticles containing isoniazid were prepared by desolvation, in which urea was used as a denaturing agent and ethanol was used as a desolvating agent, followed by a reduction step with L-cysteine [27]. Immobilization of the drug isoniazid was carried out using the inclusion method (Figure 1).

We used urea as a mild chaotropic agent to increase the availability of albumin functional groups for thiol–disulfide exchange. Chaotropic effects begin to appear after 2–3 h, so in order to speed up the reaction and provide a higher activation energy between the protein and the chaotropic agent, we used ultrasound treatment [35].

The central composite design method (CCD) was used to study the effect of the formulation factors (concentrations of HSA, urea, L-cysteine, rifampicin and isoniazid) on the dependent physico-chemical characteristics, particle size and drug loading degree. CCD is an alternative approach that investigates a large number of variables at different levels with a limited number of experiments. The variables in Table 1 have been chosen based on our initial experiments [27,34,34,36,37,38]. Table 2 presents the results of the effects of the estimated variables on the drug loading degree and average particle size.

The particle size and polydispersity (PDI) of HSA-INH-RIF NPs ranged from 134.2 ± 7.7 to 351.6 ± 7.3 nm and 0.108 ± 0.028 to 0.452 ± 0.039, respectively. The smallest particle size was observed with the NP18 formulation (134.2 ± 7.7 nm) with 40 mg/mL albumin, 5 mol/L urea, 1.25 mg/mL cysteine and 6 mg/mL drug concentration.

The zeta (ζ) potential is an important parameter that affects the stability of colloidal systems and their behavior [39,40]. It influences the electrostatic repulsion or attraction between particles, which can prevent them from aggregating [41]. The zeta potential of nanoparticles containing albumin with a concentration of 80 mg/mL had a value of less than −30 mV; this indicates the electrical stability of the system, as charged particles will repel from each other, reducing the chance of them aggregating.

The loading degree and NP yield are important parameters in the synthesis of polymeric carriers. The drug loading and NP yield ranged from 8 to 90% and 5 to 69%, respectively.

Analysis of variance (ANOVA) was applied to examine the suitability and significance of the mathematical model to estimate the particle size and loading degree (Table 3). The multiple regression data showed that the quadratic terms should be retained in the mathematical model to determine the responses.

The F-value of the model for the mean size, equal to 10.52, means that the model is significant. The probability that such a large F-value could arise due to noise is only 1.71%. A *p*-value of less than 0.0500 indicates that the model conditions are significant (Table 3).

The F-value of the model for rifampicin-loading degree, equal to 4.93, means that the model is significant. The probability that such a high F-value could arise from noise is only 4.26%.

A model F-value for isoniazid-loading degree of 4.70 means that the model is significant. The probability that such a large F-value could arise from noise is only 4.69%.

The model developed on the basis of the CCD to estimate the particle size and the degree of loading of each drug is as follows:$$\begin{array}{cc} Size=205.99 & +3.11A+13.61B+20.56C-4.45D+18.26E+23.23AB\\ & +15.30AC+49.53AD-21.70AE+14.87BC+19.98BD\\ & -52.02BE+24.58CD-47.50CE+26.04DE+48.78A^{2}\\ & -30.56B^{2}+22.84C^{2}-11.10D^{2}-8.08E^{2}\\ Loading\;degree & of\;RIF\\ & =25.02-7.69A+9.36B-6.02C+6.35D+21.40E\\ & +17.55AB+29.43AC+10.06AD-0.99AE+18.38BC\\ & -12.74BD-25.29BE+3.14CD-5.41CE-15.53DE\\ & -1.57A^{2}+2.68B^{2}-0.45C^{2}-6.49D^{2}+12.07E^{2} \end{array}$$
$$\begin{array}{cc} Loading\;degree & of\;INH\\ & =20.62-1.34A+9.36B+1.00C+6.35D+3.34E\\ & +1.83AB+6.94AC+4.34AD+7.60AE+5.49BC-10.36BD\\ & -1.35BE-2.50CD+10.26CE-4.84DE-2.23A^{2}+6.27B^{2}\\ & +7.83C^{2}-2.90D^{2}-4.47E^{2} \end{array}$$

These equations take into account only those conditions that are statistically significant, as shown by ANOVA. The relationship between the dependent and independent parameters can be represented graphically by means of response surface diagrams for the two variables simultaneously.

The influence of the different factors on the particle size has been evaluated by means of a three-dimensional (3D) response surface diagram (shown in Figure 2).

Therefore, the plots were prepared by plotting the response surface (i.e., average NPs size) against the two effective factors, which shows a binary interaction, while the other factors remain unchanged. Figure 2a shows that with increasing concentrations of HSA and urea, the average particle size increased, but at central concentrations of albumin ([HSA] = 40 mg/mL) the NP size decreased, while at central concentrations of urea ([Urea] = 5 mg/mL) the average size increased. At central concentrations of cysteine ([L-cysteine] = 1.25 mg/mL), the smallest NP size was achieved (Figure 2b). Drug concentrations (Figure 3c) had a similar effect on the average particle size, with increasing drug concentrations decreasing the average particle size; thus, using the highest concentration of isoniazid resulted in the smallest particle size (at [INH] = 10 mg/mL, the average particle size was 141.6 nm as predicted by Design Expert) compared to rifampicin (at [RIF] = 10 mg/mL, the average particle size was 185.7 nm as predicted by Design Expert).

The effect of the various factors on the loading degree of rifampicin and isoniazid was further evaluated using a 3D surface response graph (Figure 3 and Figure 4).

Figure 3a and Figure 4a show that the drug loading degree decreased with increasing albumin concentration. As the concentration of the drug increased, the loading degree increased, which is consistent. When the highest concentration of urea was used, the NPs with the highest drug loading were obtained (Figure 3b and Figure 4b). The L-cysteine concentration did not significantly affect the loading degree of rifampicin (Figure 3c) and the loading degree of isoniazid increased when the L-cysteine concentration increased (Figure 4c).

After analyzing the data using ANOVA, parameters were selected to optimize the process to produce NPs with the minimum size and maximum drug loading (Figure 5). The best parameters to obtain HSA-INH-RIF nanoparticles were found to be the following concentrations: HSA—20 mg/mL, urea—3.78 mol/L, L-cysteine—0.5 mg/mL, isoniazid—8 mg/mL and rifampicin—4 mg/mL.

Experiments were carried out to confirm the predicted optimum parameters from the CCD method. A good agreement between the predicted particle size and the experimental particle size was observed (Table 4). Hence, the size of the synthesized HSA-INH-RIF nanoparticles can be improved by the CCD method.

The use of the produced HSA-INH-RIF NPs in further studies as well as in practice implies that the nanoparticles suspension should be aggregation-stable for some time [33]. Therefore, the aggregation stability of NPs was studied over a 24 h period. The particle size and zeta potential of the HSA-INH-RIF NPs in suspension were determined by DLS and the results are presented in Table 5.

The results shown in Table 5 illustrate that the particle size increased slightly in the above time interval and did not form larger fractions and aggregates (as evidenced by both the average particle size and zeta potential).

### 3.2. Study of Physico-Chemical Parameters of the HSA-INH-RIF Nanoparticles

The morphology of the produced HSA-INH-RIF nanoparticles was studied by scanning electron microscopy (SEM) and the images obtained are shown in Figure 6. The particles had a spherical morphology and the average size of the nanoparticles calculated using ImageJ 1.53e software was 188.7 ± 8.8 nm.

In order to confirm the incorporation of the drugs into the polymer matrix, differential scanning calorimetry (DSC) and thermogravimetric analysis (TGA) were performed; the thermograms of isoniazid, rifampicin, HSA NPs and HSA-INH-RIF NPs are shown in Figure 7.

The main endothermic peak for INH and RIF (pure preparations) was observed at 177.36 and 258.75 °C, respectively (Figure 7). The sharply tapering peak indicates a crystalline form of the drugs. A mass loss of up to 68% of isoniazid occurred between 260 and 450 °C, which may correspond to the decomposition temperature of the drug [42]. The presence of RIF as polymorph I can be explained by the TGA thermogram (Figure 7b). The thermal decomposition process occurred in two stages [43]. The first stage occurred rapidly between 231 and 283 °C with a mass loss of about 21.5%, while the second stage occurred more slowly, between 292 and 615 °C with a mass loss of 50%. The HSA NPs showed a characteristic endothermic peak at around 120 °C, which probably corresponds to their melting period (Figure 7c). The DSC for the HSA-INH-RIF NPs showed a slight shift of the endothermic peak towards higher temperatures compared to empty albumin NPs, probably due to the inclusion of drugs in the polymer matrix (Figure 7d).

To confirm the inclusion of drugs in the HSA NPs complex, IR spectroscopy of the standard components as controls as well as the obtained NPs was carried out. Figure 8 shows the IR spectra of isoniazid, rifampicin, empty HSA NPs and nanoparticles immobilized with anti-TB drugs.

Isoniazid (Figure 8) exhibited characteristic absorption bands: the -NH_2_ bond at 3306 cm^−1^, the C=O bond with the pyridine fragment at 1666 cm^−1^, the -C-N bond at 1550 cm^−1^, and the -N-N amide group at 1140 cm^−1^ [27,44]. The FT-IR spectrum of rifampicin exhibited characteristic peaks at different wave numbers: 3476 cm^−1^ is related to the NH stretch, 2889 cm^−1^ is related to the C-H bond, 1728 cm^−1^ corresponds to the C=O bond, 1462 cm^−1^ indicates C=C, 1365 cm^−1^ is related to CH_2_ and C=C, and 1060 cm^−1^ is related to -CH, CO and C-H [45,46].

The presented spectra contain albumin characteristic bands detected at specific wave numbers. In particular, the following bands were observed [33]: the band detected at 3430 cm^−1^ corresponds to the A-amide group bound to N-H; the pointed peak at 2920 cm^−1^ corresponds to the B-amide group bound to the free ion; the peak recorded at 1535 cm^−1^ corresponds to amide II and is related to C-N stretching and N-H bending; the peak at 1610 cm^−1^ corresponds to amide I and is related to the C-O bond; the CH_2_ groups are located at 1396 cm^−1^; and the peak of amide III, detected at about 1245 cm^−1^, is related to C-N group stretching and N-H bending.

The FTIR spectrum related to the nanoparticles was similar to the polymer spectrum. In the IR spectra of HSA-INH-RIF nanoparticles, there are peaks at 3439 cm^−1^ for OH groups and 1610 cm^−1^ for C=O bonds. As the FTIR spectra of the nanoparticles showed no clear and optimal peak characteristics for rifampicin and isoniazid, which can be hidden by bonds formed by the polymers, it can be concluded that the drug immobilization into nanoparticles was successful [45].

The in vitro release of rifampicin and isoniazid from the polymeric albumin matrix was studied using the dialysis method (Figure 9).

HSA-INH NPs produced under optimized conditions (HSA—20 mg/mL, urea—3.78 mol/L, L-cysteine—0.5 mg/mL, isoniazid—8 mg/mL, rifampicin—4 mg/mL) were dispersed in phosphate-buffered saline (PBS; pH = 7.4) at 200 rpm and 37 °C. The amount of RIF and INH released into the medium was determined at 0.25, 0.5, 1, 2, 4, 8, 24 h intervals using HPLC (Appendix A) [47,48]. A slow and prolonged release of the immobilized drugs was observed. In the first 2 h, 2.9 mg/mL rifampicin and 3.7 mg/mL isoniazid were released from the nanoparticles, after which a gradual and sustained release to 4.3 mg/mL rifampicin and 5.2 mg/mL isoniazid was observed after 24 h. The initial burst of release could be due to the release of the drug adsorbed on the surface of the nanoparticles [27]. Release then occurs by diffusion of drug molecules through the polymer matrix of the nanoparticles [49]. The data in Figure 9 show that the release is prolonged, preventing fluctuations in blood concentrations and maintaining a constant therapeutic concentration for an extended period of time.

## 4. Conclusions

Optimization of albumin-based nanoparticles loaded with rifampicin and isoniazid by desolvation is a promising approach to improve the properties of anti-TB drugs. The use of nanoparticles achieves multiple advantages such as increased stability, controlled size and shape, as well as increased bioavailability and drug activity. The central composite design (CCD) method was used to study the effect of formulation factors (concentrations of HSA, urea, L-cysteine, rifampicin and isoniazid) on the dependent physico-chemical characteristics, particle size and drug loading degree. The obtained nanoparticles had satisfactory characteristics (mean size 216.7 ± 3.7 nm, polydispersity 0.286 ± 4.9, loading degree of rifampicin 44% and isoniazid 27% and NP yield 45%) and spherical shape. The results are in good agreement with those analyzed using the CCD method. The method of central composite design provides an efficient approach to optimize nanoparticle synthesis. The effects of different factors on particle size, polydispersity index and drug loading degree were analyzed by ANOVA. The physical and chemical characteristics of the nanoparticles were determined by SEM, TGA, DSC and IR spectroscopy. The in vitro results showed that the release of isoniazid and rifampicin from the albumin nanoparticles is prolonged. Further toxicological and antimycobacterial studies will be carried out to evaluate their safety and efficacy at different stages of clinical trials.

## Figures and Tables

**Figure 1 polymers-15-02774-f001:**
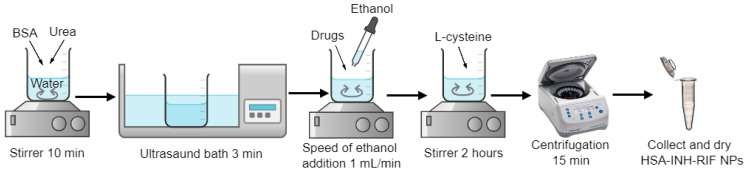
Scheme for producing HSA-INH-RIF nanoparticles.

**Figure 2 polymers-15-02774-f002:**
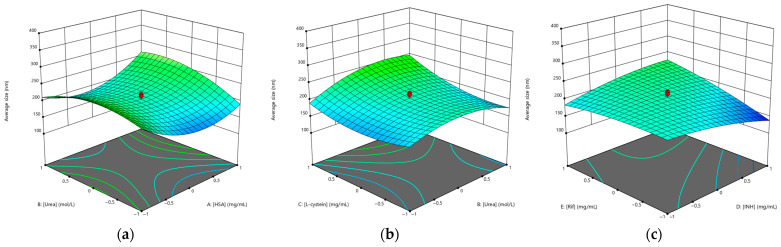
Three-dimensional (3D) response surface plots of the effect of independent factors on average particle size: (**a**) [HSA]–[urea]; (**b**) [urea]–[L-cysteine]; (**c**) [INH]–[RIF].

**Figure 3 polymers-15-02774-f003:**
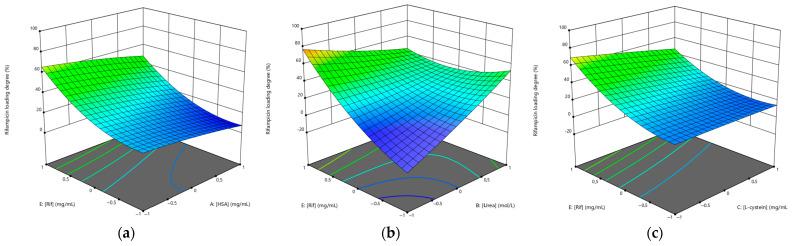
Three-dimensional (3D) response surface plots of the effect of independent factors on the loading degree of rifampicin: (**a**) [HSA]–[RIF]; (**b**) [urea]–[RIF]; (**c**) [L-cysteine]–[RIF].

**Figure 4 polymers-15-02774-f004:**
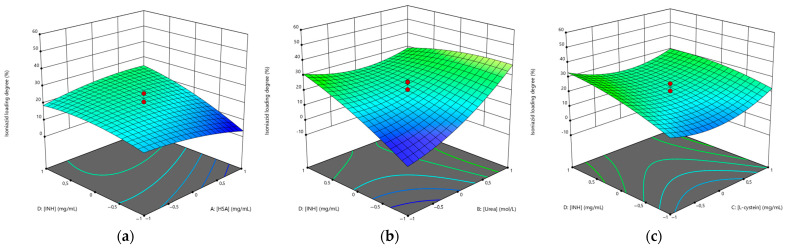
Three-dimensional (3D) response surface plots of the effect of independent factors on isoniazid loading degree: (**a**) [HSA]–[INH]; (**b**) [urea]–[INH]; (**c**) [L-cysteine]–[INH].

**Figure 5 polymers-15-02774-f005:**
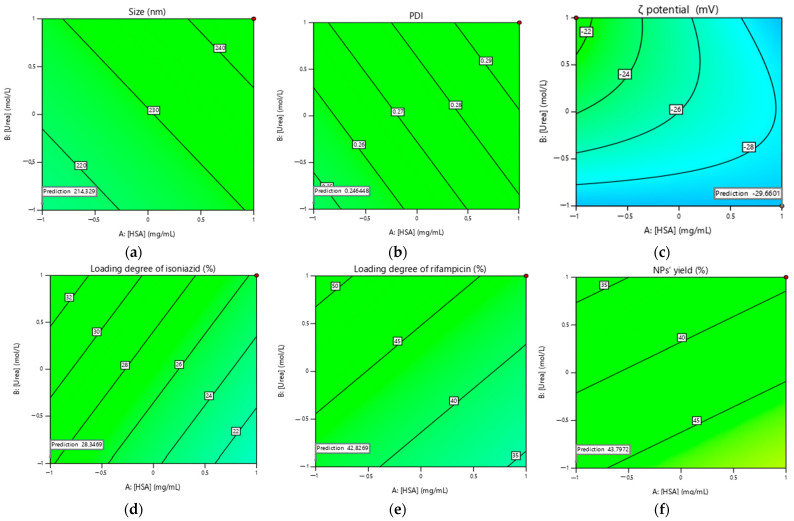
Contour plots of the best parameters to obtain HSA-INH-RIF nanoparticles based on (**a**) particle size, (**b**) polydispersity index, (**c**) ζ potential, (**d**) loading degree of isoniazid, (**e**) loading degree of rifampicin and (**f**) NP yield.

**Figure 6 polymers-15-02774-f006:**
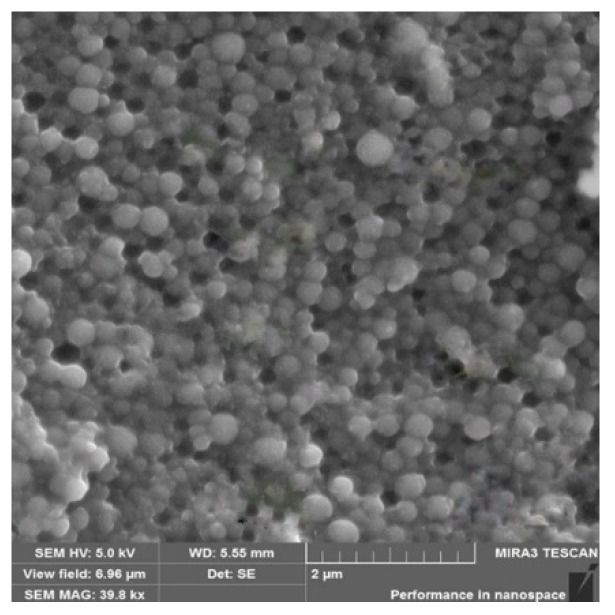
SEM image of optimized HSA-INH-RIF nanoparticles.

**Figure 7 polymers-15-02774-f007:**
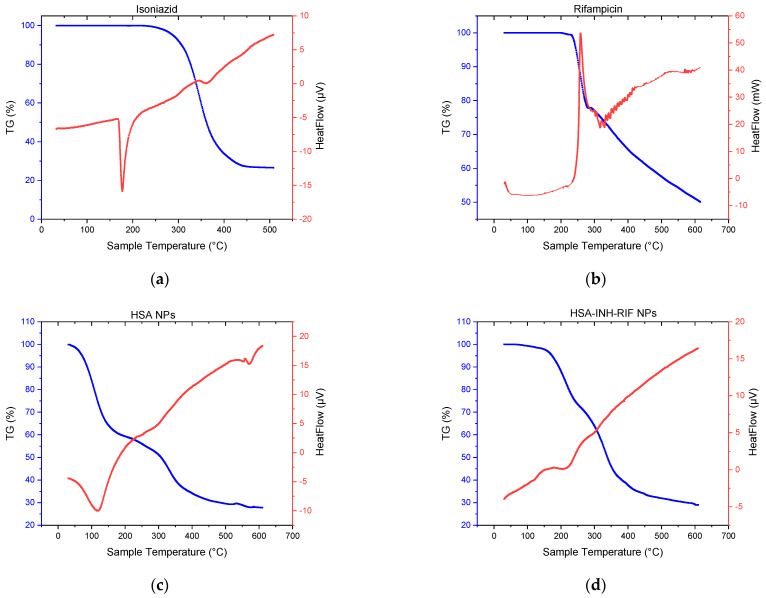
Thermogravimetric analysis and differential scanning calorimetry of (**a**) Isoniazid, (**b**) Rifampicin, (**c**) HSA NPs, (**d**) HSA-INH-RIF NPs.

**Figure 8 polymers-15-02774-f008:**
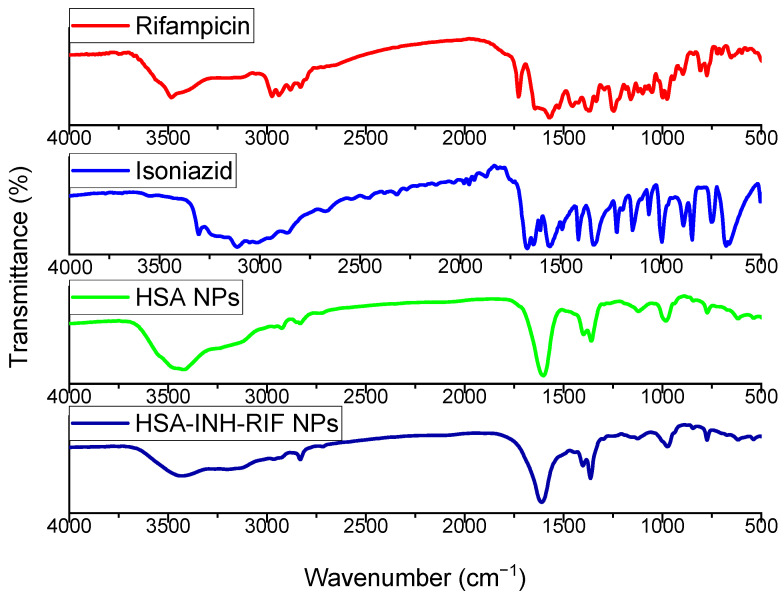
Infrared spectra of the anti-TB drugs isoniazid and rifampicin, empty HSA NPs and HSA-INH-RIF NPs.

**Figure 9 polymers-15-02774-f009:**
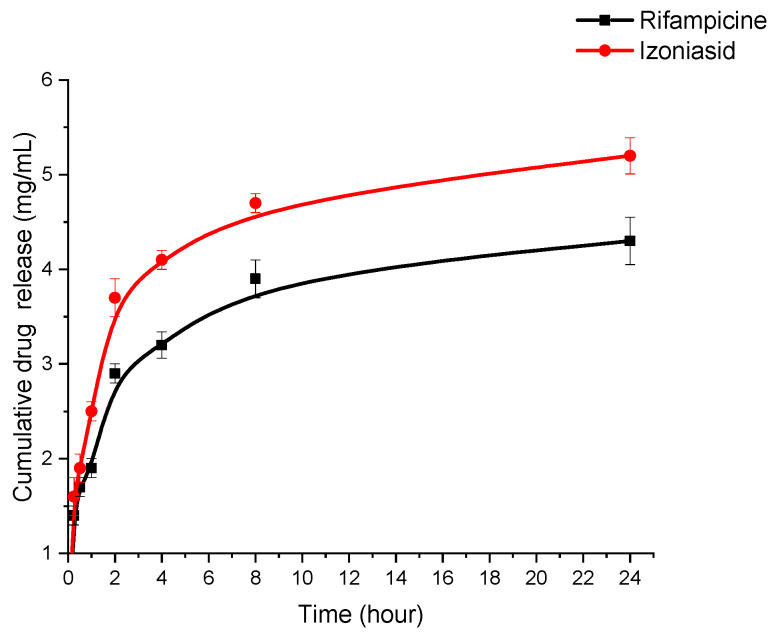
Cumulative release of rifampicin and isoniazid from the polymeric matrix of HSA nanoparticles.

**Table 1 polymers-15-02774-t001:** Experimental factors for synthesis of HSA-INH-RIF NPs and corresponding levels.

Independent Variable	Measuring Unit	Variable Levels				
		Star-Low (−1.82)	Low (−1)	Center (0)	High (1)	Star-High (1.82)
A: [HSA]	mg/mL	10	20	40	80	100
B: [Urea]	mol/L	3	4	5	6	7
C: [L-cysteine]	mg/mL	0.1	0.5	1.25	2	2.5
D: [INH]	mg/mL	2	4	6	8	10
E: [Rif]	mg/mL	2	4	6	8	10

**Table 2 polymers-15-02774-t002:** Formulations of HSA-INH-RIF NPs using central composite design and their evaluation parameters.

Formulation Code	A: [HSA] (mg/mL)	B: [Urea] (mol/L)	C: [L-Cysteine] (mg/mL)	D: [INH] (mg/mL)	E: [RIF] (mg/mL)	Size (nm)	PDI	ζ Potential (mV)	Loading Degree of INH (%)	Loading Degree of RIF (%)	NP Yield (%)
NP1	80	4	2	8	4	304.7 ± 9.1	0.108 ± 0.028	−30.0 ± 1.6	17	17	66
NP2	40	5	1.25	6	10	351.6 ± 7.3	0.146 ± 0.003	−14.9 ± 2.4	21	25	21
NP3	80	6	2	4	4	204.7 ± 4.1	0.135 ± 0.037	−31.1 ± 1.6	15	7	54
NP4	40	5	1.25	6	6	191.4 ± 9.4	0.208 ± 0.009	−19.7 ± 1.8	6	12	66
NP5	40	5	2.5	6	6	239.6 ± 8.8	0.452 ± 0.039	−14.8 ± 2.6	30	51	34
NP6	100	5	1.25	6	6	184.2 ± 9.6	0.446 ± 0.004	−18.5 ± 2.6	20	39	6
NP7	40	5	1.25	6	6	336.6 ± 9.5	0.345 ± 0.048	−20.0 ± 1.1	53	32	33
NP8	40	5	0.1	6	6	156.1 ± 8.4	0.198 ± 0.043	−24.7 ± 1.4	7	13	68
NP9	80	6	0.5	8	4	231.4 ± 1.8	0.207 ± 0.049	−34.1 ± 1.5	22	22	12
NP10	40	5	1.25	2	6	177.5 ± 5.3	0.171 ± 0.039	−22.9 ± 1.7	51	51	27
NP11	40	5	1.25	6	10	289.5 ± 2.9	0.145 ± 0.004	−24.4 ± 1.8	25	78	5
NP12	20	4	2	8	8	233.2 ± 5.6	0.191 ± 0.047	−19.6 ± 2.2	26	26	43
NP13	80	4	2	4	8	191.6 ± 9.7	0.335 ± 0.016	−35.2 ± 1.6	21	21	53
NP14	20	6	2	8	4	232.1 ± 3.9	0.424 ± 0.006	−22.7 ± 1.1	18	90	14
NP15	80	4	0.5	8	8	215.5 ± 2.4	0.414 ± 0.075	−32.1 ± 1.5	32	33	69
NP16	40	7	1.25	6	6	136.3 ± 9.7	0.189 ± 0.038	−11.2 ± 2.1	23	24	39
NP17	40	5	1.25	6	6	216.5 ± 5.4	0.241 ± 0.025	−22.6 ± 1.4	34	21	51
NP18	40	5	1.25	6	6	134.2 ± 7.7	0.259 ± 0.001	−22.6 ± 1.9	23	25	49
NP19	20	6	0.5	8	8	243.2 ± 5.8	0.193 ± 0.005	−16.0 ± 1.6	29	22	7
NP20	40	5	1.25	6	6	134.2 ± 7.7	0.127 ± 0.030	−18.3 ± 2.6	19	30	38
NP21	10	5	1.25	6	6	253.0 ± 5.7	0.248 ± 0.019	−6.2 ± 0.7	39	39	15
NP22	40	3	1.25	6	6	221.2 ± 4.4	0.443 ± 0.056	−18.4 ± 2.2	26	23	36
NP23	20	6	2	4	8	327.3 ± 6.6	0.347 ± 0.002	−22.7 ± 1.1	42	21	20
NP24	40	5	1.25	10	6	152.8 ± 6.5	0.187 ± 0.016	−11.2 ± 3.7	20	19	26
NP25	20	4	0.5	4	4	174.9 ± 2.9	0.214 ± 0.007	−22.7 ± 1.3	7	7	63
NP26	80	6	0.5	4	8	159.6 ± 2.8	0.189 ± 0.032	−19.3 ± 2.2	8	26	45

**Table 3 polymers-15-02774-t003:** ANOVA results for particles size and drug-loading efficiency.

Response	Source	Sum of Squares	Degree of Freedom	Mean Square	F-Value	*p*-Value	
Size	Model	8.643 × 10^12^	21	4.116 × 10^11^	10.52	0.0171	significant
	Pure Error	1.564 × 10^11^	4	3.911 × 10^10^			
	Residual	1.564 × 10^11^	4	3.911 × 10^10^			
	Lack of Fit	5.223 × 10^11^	1	5.223 × 10^11^	13.36	0.0217	
	Cor Total	8.800 × 10^12^	25				
Loading degree of INH	Model	6.619 × 10^8^	20	3.310 × 10^7^	4.70	0.0469	significant
	Pure Error	6.019 × 10^6^	4	1.505 × 10^6^			
	Residual	6.019 × 10^6^	4	1.505 × 10^6^			
	Lack of Fit	2.917 × 10^7^	1	2.917 × 10^7^	19.39	0.0117	
	Cor Total	6.971 × 10^8^	25				
Loading degree of RIF	Model	7.385 × 10^9^	2	3.692 × 10^8^	4.93	0.0426	significant
	Pure Error	2.429 × 10^6^	4	6.072 × 10^5^			
	Residual	2.429 × 10^6^	4	6.072 × 10^5^			
	Lack of Fit	3.722 × 10^8^	1	3.722 × 10^8^	612.97	<0.0001	
	Cor Total	7.759 × 10^9^	25				

**Table 4 polymers-15-02774-t004:** Predicted and experimental results for HSA-INH-RIF NPs.

	Size (nm)	PDI	ζ Potential (mV)	Loading Degree of Rifampicin, %	Loading Degree of Isoniazid, %	NP Yield, %
Predicted	214.2	0.246	−29.6	43	29	44
Experimental	216.7 ± 3.7	0.286 ± 4.9	−26.7 ± 1.5	44	27	45
Error %	1.2	16	11	2.3	6.9	2.3

**Table 5 polymers-15-02774-t005:** The stability of produced HSA-INH-RIF NPs.

Time	Size (nm)	ζ Potential (mV)
Immediately after purification	216.7 ± 3.7	−26.7 ± 1.5
After 1 h	217.7 ± 2.4	−25.9 ± 1.9
After 2 h	219.4 ± 2.1	−26.1 ± 1.4
After 4 h	227.1 ± 2.7	−26.4 ± 1.2
After 8 h	236.9 ± 3.1	−26.5 ± 1.7
After 1 day	242.4 ± 5.7	−27.3 ± 1.1

## Data Availability

The data presented in this study are available on request from the corresponding author.

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
