# Peer review of "Human Serum Albumin Nanoparticles: Synthesis, Optimization and Immobilization with Antituberculosis Drugs"

_polymers, 2023, doi:10.3390/polym15132774_

Round 1
Reviewer 1 Report
Dear Authors
The article entitled “Human serum albumin nanoparticles: synthesis, optimization and immobilization with antituberculosis drugs” is a very interesting topic but very poorly designed. Fallowing are few suggestions which might be helpful for the improvement of the article a bit
· What are the benefits of albumin-based nanoparticles over polylactide, copolymers of polylactic and glycolic acid nanoparticles? The authors should elaborate on the relevance of albumin nanoparticles.
· Why use the DoE's CCD model instead of another models? The article lack novelty.
· Authors should include a novelty statement in the introductory part to make their work more appealing and attractive to the reader.
· What role does ultrasound play in the desolvation process?
· What solvent is used to make the drugs solution?
· The authors inserted two drugs, isoniazid (INH) and rifampicin (RIF). Do the authors did the simultaneous estimation of the drugs for the loading efficacy and release.? If so, the authors should give the HPLC method development. Why did the authors use a temperature of 40 °C for the HPLC study?
· In accordance with the process If ultrasonic is so important in the formulation, why don't the authors include it as an independent element in the CCD model?
· On which criteria did the authors choose the low and high concentrations of independent variables? Which criteria are used to pick the star low and high value?
· Figure 1 should be provided in high resolution by the authors.
· The FTIR should be provided by the authors in order to understand the compatibility of the components and the final formulation.
· In Table 3, authors should provide the Residual, Lack of Fit, Pure Error, and Corr. Total value.
· Authors should provide contour plots to support the NP18 formulation.
· Authors should provide detailed explanations for why they chose NP18.
· Authors should include the DSC and TGA sections in the methods section as well.
· The authors claim that the loaded drugs amount in NP18 is 8 mg/mL, with a loading degree of 23 and 25% (table 2). The authors obtained a medication release of between 4.3 and 5.2 mg/mL. For a better comprehension, authors should describe the results with sufficient rationale.
· “The amount of drug released into the medium was determined at 24-hour intervals by HPLC.” How authors got the intermediate values?
· Rewrite the conclusion with the title claim.
The article needs attention in English grammar and sentence construction.
Author Response
Thank you for your review of our paper. We have answered each of your points below.
1) What are the benefits of albumin-based nanoparticles over polylactide, copolymers of polylactic and glycolic acid nanoparticles? The authors should elaborate on the relevance of albumin nanoparticles.
Response: We agree with the comment. We have added about the benefits of albumin
2) Why use the DoE's CCD model instead of another models? The article lack novelty
Response: We agree with the comment. We have previously looked at the Taguchi method for optimizing NPs, so it was interesting to use the CCD method. We agree with your comment, have added novelty to the introduction
3) Authors should include a novelty statement in the introductory part to make their work more appealing and attractive to the reader
Response: We agree with your comment, have added novelty to the introduction
4) What role does ultrasound play in the desolvation process?
Response: Ultrasound speeds up chemical reactions by providing a higher activation energy between albumin and urea. We have added a reference to the literature in the extraction methodology.
5) What solvent is used to make the drugs solution?
Response: We agree with your comment, we have added a solvent which is used to prepare a drug solution
6) The authors inserted two drugs, isoniazid (INH) and rifampicin (RIF). Do the authors did the simultaneous estimation of the drugs for the loading efficacy and release.? If so, the authors should give the HPLC method development. Why did the authors use a temperature of 40 °C for the HPLC study?
Response: We agree with your comment, the development of the HPLC method has been added.
In order to keep the blurring of the peaks to a minimum, the column temperature has to be above room temperature. Thus at 40 °C more accurate analyses are obtained
7) In accordance with the process If ultrasonic is so important in the formulation, why don't the authors include it as an independent element in the CCD model?
Response: We agree with your comment. Indeed, the reviewer is right that this parameter affects the nanoparticle production process, it refers to an external factor, but in this paper we have not considered the influence of external factors
8) On which criteria did the authors choose the low and high concentrations of independent variables? Which criteria are used to pick the star low and high value?
Response: Earlier studies were conducted for albumin nanoparticles and considered different concentrations of independent variables, and based on previously obtained data, the necessary parameters were selected.
9) Figure 1 should be provided in high resolution by the authors.
Response: We agree with your comment. we replaced the Figure 1
10) The FTIR should be provided by the authors in order to understand the compatibility of the components and the final formulation.
Response: We agree with your comment. We've added an IR spectrum
11) In Table 3, authors should provide the Residual, Lack of Fit, Pure Error, and Corr. Total value.
Response: We agree with your comment. We've added добавили Residual, Lack of Fit, Pure Error, and Corr. Total value.
12) Authors should provide contour plots to support the NP18 formulation.
Response: In the discussion it was pointed out that NP18 had the smallest size of the data obtained, but this does not mean that NP18 was chosen as the optimized formulation.
After processing the data by ANOVA, parameters were selected to optimize the process to produce NPs with minimum size and maximum drug loading. The best pa-rameters to obtain HSA-INH-RIF nanoparticles were found to be the following con-centrations: HSA – 20 mg/mL, Urea – 3.78 mol/L, L-cysteine – 0.5 mg/mL, isoniazid – 8 mg/mL, rifampicin – 4mg/mL. Experiments were carried out to confirm the predicted optimum parameters from the CCD method.
13) Authors should provide detailed explanations for why they chose NP18.
Response: After processing the data by ANOVA, parameters were selected to optimize the process to produce NPs with minimum size and maximum drug loading. The best pa-rameters to obtain HSA-INH-RIF nanoparticles were found to be the following con-centrations: HSA – 20 mg/mL, Urea – 3.78 mol/L, L-cysteine – 0.5 mg/mL, isoniazid – 8 mg/mL, rifampicin – 4mg/mL. Experiments were carried out to confirm the predicted optimum parameters from the CCD method.
14) Authors should include the DSC and TGA sections in the methods section as well.
Response: We agree with your comment. We have added the DSC and TGA methodology
15) The authors claim that the loaded drugs amount in NP18 is 8 mg/mL, with a loading degree of 23 and 25% (table 2). The authors obtained a medication release of between 4.3 and 5.2 mg/mL. For a better comprehension, authors should describe the results with sufficient rationale.
Response: To study the degree of release, NÐ s under optimized conditions (HSA 20 mg/mL, Urea 3.78 mol/L, L-cysteine 0.5 mg/mL, isoniazid 8 mg/mL, rifampicin 4mg/mL) were used, where the loading degrees of rifampicin and isoniazid were 44 and 27% respectively. We added to the discussion on investigating the release of drugs from the polymer matrix
16) “The amount of drug released into the medium was determined at 24-hour intervals by HPLC.” How authors got the intermediate values?
Response: We agree with your comment and have added: Dialysate samples were taken at fixed time intervals such as 0.25, 0.5, 1, 2, 4, 8, 24 hours
17) Rewrite the conclusion with the title claim.
Response: We agree with your comment. We have supplemented the conclusion
Reviewer 2 Report
The article «Human serum albumin nanoparticles: synthesis, optimization 2 and immobilization with antituberculosis drugs» is attracted to a relevant topic and has a high applied value.
The aim of this article was to create nanoparticles of human serum albumin immobilized with antituberculosis drugs (rifampicin, isoniazid) using the desolvation method. The loading degree of rifampicin and isoniazid in the optimized nanoparticles were 44% and 27% respectively. The drug release from the polymer matrix was studied using dialysis membranes.
The article is well structured, written in sufficient detail and logically.
However, a number of serious questions arise:
1. It is not clear why the authors call particles with average diameter of 216.7±3.7 nm nanoparticles, in my opinion the term microparticles is more correct.
2. The authors do not provide any information about the stability of their particles over time. As a minimum, they can provide data on the change in particle size and zeta potential at regular intervals (1-2 hours) of time within 24 hours.
3. It would be nice to supplement the article with experiments to determine the toxicity of particles and their effectiveness in relation to Mycobacterium tuberculosis, comparing these parameters for particles developed by the authors with those for rifampicin and isoniazid in terms of the concentration of the antibiotics in the system. Without these experiments, the title of the article is not justified and the work loses its scientific relevance.
4. 8 out of 30 references (that is, almost 30%) are the works of the author Tazhbayev. None of these references are in the Materials and Methods section. I think that the authors are too fond of self-quoting. some references to their own work should be removed, or their justification should be proved.
Minor editing of English language required
Author Response
Thank you for your comments. Our answers to your points are as follows.
1) It is not clear why the authors call particles with average diameter of 216.7±3.7 nm nanoparticles, in my opinion the term microparticles is more correct.
Response: Nanoparticles by a definition given by Kreuter J. (which was later adopted by the Encyclopedia of Pharmaceutical Technology and the Encyclopedia of Nanotechnology) “are solid colloidal particles ranging in size from 10 to 1000 nm (1 μm). They consist of macromolecular material in which the active principle (drug or biologically active material) is dissolved, entrapped, encapsulated and/or to which the active principle is adsorbed or “attached” [Kreuter. J. Colloidal Drug Delivery Systems.-New York: Marcel Dekker, 1994.- Ð .344.].
2) The authors do not provide any information about the stability of their particles over time. As a minimum, they can provide data on the change in particle size and zeta potential at regular intervals (1-2 hours) of time within 24 hours.
Response: We agree with your comment, we have added the results of the zeta potential study.
3) It would be nice to supplement the article with experiments to determine the toxicity of particles and their effectiveness in relation to Mycobacterium tuberculosis, comparing these parameters for particles developed by the authors with those for rifampicin and isoniazid in terms of the concentration of the antibiotics in the system. Without these experiments, the title of the article is not justified and the work loses its scientific relevance.
Response: A study is currently being carried out at a special institution to inhibit mycobacterial growth, the preliminary results are good, but it takes quite a long study, which will be published in Part 2 as a stand-alone study.
4) 8 out of 30 references (that is, almost 30%) are the works of the author Tazhbayev. None of these references are in the Materials and Methods section. I think that the authors are too fond of self-quoting. some references to their own work should be removed, or their justification should be proved.
Response: We agree with your comment, we have refined the list of references
Round 2
Reviewer 1 Report
Dear Authors
The article entitled “Human serum albumin nanoparticles: synthesis, optimization and immobilization with antituberculosis drugs” is well revised. Fallowing are few suggestions which might be helpful for the improvement of the article a bit
· Add the exclusion criteria of the ultrasound in the article
· Do the authors employ the simultaneous estimation HPLC technique to estimate isoniazid (INH) and rifampicin (RIF) together? If yes, provide the HPLC technique development for the simultaneous quantification of both drugs. If not, how did the authors estimate two medicines during the in vitro release investigation from the dual drug loaded formulation?
· The authors said that essential parameters were chosen based on previous research of albumin nanoparticles and varied concentrations of independent factors. Please offer information from previous research with reference for a better understanding. Authors can use DoE to pick primary parameter, check the fallowing articles
· Implementation of two different experimental designs for screening and optimization of process parameters for metformin-loaded carboxymethyl chitosan formulation
· Statistical Design for Formulation Optimization of Hydrocortisone Butyrate-Loaded PLGA Nanoparticles
· If NP18 is not chosen for the optimized formulation, whose formulation was the best batch from the CDD runs? Please provide contour plots for the same batch.
The articles need minor English correction and line construction.
Author Response
Thank you for your review of our paper. We have answered each of your points below.
- Add the exclusion criteria of the ultrasound in the article
We agree with your comment, added to the article about the role of ultrasound
- Do the authors employ the simultaneous estimation HPLC technique to estimate isoniazid (INH) and rifampicin (RIF) together? If yes, provide the HPLC technique development for the simultaneous quantification of both drugs. If not, how did the authors estimate two medicines during the in vitro release investigation from the dual drug loaded formulation?
We agree with your comment and have added HPLC curves to the supplementary material. We determined isoniazid (INH) and rifampicin simultaneously by HPLC. HPLC chromatograms show separation for isoniazid and rifampicin. Each drug leaves the HPLC column at a specific time.
- The authors said that essential parameters were chosen based on previous research of albumin nanoparticles and varied concentrations of independent factors. Please offer information from previous research with reference for a better understanding. Authors can use DoE to pick primary parameter, check the fallowing articles
- Implementation of two different experimental designs for screening and optimization of process parameters for metformin-loaded carboxymethyl chitosan formulation
- Statistical Design for Formulation Optimization of Hydrocortisone Butyrate-Loaded PLGA Nanoparticles
We agree with your comment and have added references to the literature
- If NP18 is not chosen for the optimized formulation, whose formulation was the best batch from the CDD runs? Please provide contour plots for the same batch.
We agree with your comment, have added contour graphics for an optimized batch
We hope the revised version is will suitable for publication.
Reviewer 2 Report
The authors made some corrections to the text of the article, but did not answer all of my comments.
Remaining remarks:
1. Nanoparticles are considered particles, one of the dimensions of which - length, height or width does not exceed 100 nm, according to Vert, M.; Doi, Y.; Hellwich, K. H.; Hess, M.; Hodge, P.; Kubisa, P.; Rinaudo, M.; Schue, F. O. (2012). "Terminology for biorelated polymers and applications (IUPAC Recommendations 2012)". Pure and Applied Chemistry. 84 (2): 377 410. doi:10.1351/PAC-REC-10-12-04. S2CID 98107080. The authors argue otherwise.
2. The authors measured the zeta potential of their particles, however, they should have done this more than once in dynamics at regular intervals, as I wrote in the previous remark:
“The authors do not provide any information about the stability of their particles over time. As a minimum, they can provide data on the change in particle size and zeta potential at regular intervals (1-2 hours) of time within 24 hours”.
3. The authors write that with Mycobacterium tuberculosis the preliminary results with Mycobacterium tuberculosis are good and that they will publish them in a future article. I think that it is not worth dividing the material into two articles, but to make one, but a good one, corresponding to a high-level journal
Minor editing of English language required
Author Response
Thank you for your comments. Our answers to your points are as follows
- Nanoparticles are considered particles, one of the dimensions of which - length, height or width does not exceed 100 nm, according to Vert, M.; Doi, Y.; Hellwich, K. H.; Hess, M.; Hodge, P.; Kubisa, P.; Rinaudo, M.; Schue, F. O. (2012). "Terminology for biorelated polymers and applications (IUPAC Recommendations 2012)". Pure and Applied Chemistry. 84 (2): 377 410. doi:10.1351/PAC-REC-10-12-04. S2CID 98107080. The authors argue otherwise.
According to Vert, M.; Doi, Y.; Hellwich, K. H.; Hess, M.; Hodge, P.; Kubisa, P.; Rinaudo, M.; Schue, F. O. (2012). "Terminology for biorelated polymers and applications (IUPAC Recommendations 2012)". Pure and Applied Chemistry. 84 (2): 377 410. doi:10.1351/PAC-REC-10-12-04. S2CID 98107080: Note 3: Because other phenomena (transparency or turbidity, ultrafiltration, stable dispersion, etc.) are occasionally considered that extend the upper limit, the use of the prefix “nano” is accepted for dimensions smaller than 500 nm, provided reference to the definition is indicated.
Also according to Kreuter J. (which was later adopted by the Encyclopedia of Pharmaceutical Technology and the Encyclopedia of Nanotechnology) “Nanoparticles are solid colloidal particles ranging in size from 10 to 1000 nm (1 μm). They consist of macromolecular material in which the active principle (drug or biologically active material) is dissolved, entrapped, encapsulated and/or to which the active principle is adsorbed or “attached” [Kreuter. J. Colloidal Drug Delivery Systems.-New York: Marcel Dekker, 1994.- Ð .344.].
- The authors measured the zeta potential of their particles, however, they should have done this more than once in dynamics at regular intervals, as I wrote in the previous remark:
“The authors do not provide any information about the stability of their particles over time. As a minimum, they can provide data on the change in particle size and zeta potential at regular intervals (1-2 hours) of time within 24 hours”.
We agree with your comment, added study on stability
- The authors write that with Mycobacterium tuberculosisthe preliminary results with Mycobacterium tuberculosisare good and that they will publish them in a future article. I think that it is not worth dividing the material into two articles, but to make one, but a good one, corresponding to a high-level journal
We agree with the reviewer's comments, however, we would like as authors to split the article into 2 parts: the first one is presented in this manuscript and deals with issues concerning the synthesis of polymer nanoparticles and corresponds to the topic of the journal Polymers. We have initiated research with microbiologists in the field of microbiology related to this article, but the next article will focus on the bioavailability of the TB drug and the results will be published in a journal of the respective field.
We hope the revised version is will suitable for publication.
Round 3
Reviewer 1 Report
Dear Authors
The article is well revised .
Take care of the grammar and spelling before final submission
Minor grammar editing is needed
Reviewer 2 Report
The authors have corrected the comments. Article can be published
Minor editing of English language required